# Evaluating the Impact of High Intensity Interval Training on Axial Psoriatic Arthritis Based on MR Images

**DOI:** 10.3390/diagnostics12061420

**Published:** 2022-06-08

**Authors:** Ioanna Chronaiou, Guro Fanneløb Giskeødegård, Ales Neubert, Tamara Viola Hoffmann-Skjøstad, Ruth Stoklund Thomsen, Mari Hoff, Tone Frost Bathen, Beathe Sitter

**Affiliations:** 1Department of Circulation and Medical Imaging, Faculty of Medicine and Health Sciences, NTNU—Norwegian University of Science and Technology, 7491 Trondheim, Norway; ioanna.chronaiou@ntnu.no (I.C.); tone.f.bathen@ntnu.no (T.F.B.); 2Department of Public Health and Nursing, Faculty of Medicine and Health Sciences, NTNU—Norwegian University of Science and Technology, 7491 Trondheim, Norway; guro.giskeodegard@ntnu.no; 3The Australian E-Health Research Centre, CSIRO Health & Biosecurity, Brisbane 4029, Australia; ales.neubert@uqconnect.edu.au; 4Department of Radiology, St. Olavs Hospital, Trondheim University Hospital, 7006 Trondheim, Norway; tamara.viola.hoffmann-skjostad@stolav.no; 5Department of Neuromedicine and Movement Science, Faculty of Medicine and Health Sciences, NTNU—Norwegian University of Science and Technology, 7491 Trondheim, Norway; ruth.thomsen@ntnu.no (R.S.T.); mari.hoff@ntnu.no (M.H.); 6Department of Rheumatology, St. Olavs Hospital, Trondheim University Hospital, 7006 Trondheim, Norway

**Keywords:** exercise, spondyloarthropathies, psoriatic arthritis, magnetic resonance imaging, computer assisted image processing

## Abstract

High intensity interval training (HIIT) has been shown to benefit patients with psoriatic arthritis (PsA). However, magnetic resonance (MR) imaging has uncovered bone marrow edema (BME) in healthy volunteers after vigorous exercise. The purpose of this study was to investigate MR images of the spine of PsA patients for changes in BME after HIIT. PsA patients went through 11 weeks of HIIT (N = 19, 4 men, median age 52 years) or no change in physical exercise habits (N = 20, 8 men, median age 45 years). We acquired scores for joint affection and pain and short tau inversion recovery (STIR) and T1-weighted MR images of the spine at baseline and after 11 weeks. MR images were evaluated for BME by a trained radiologist, by SpondyloArthritis Research Consortium of Canada (SPARCC) scoring, and by extraction of textural features. No significant changes of BME were detected in MR images of the spine after HIIT. This was consistent for MR image evaluation by a radiologist, by SPARCC, and by texture analysis. Values of textural features were significantly different in BME compared to healthy bone marrow. In conclusion, BME in spine was not changed after HIIT, supporting that HIIT is safe for PsA patients.

## 1. Introduction

Psoriatic arthritis (PsA) is a chronic inflammatory joint disease associated with skin psoriasis that can manifest in the axial skeleton and peripheral joints, and can include dactylitis and enthesitis [1]. It is systemic inflammation associated with several comorbidities and increased cardiovascular mortality and morbidity [2,3]. The prevalence of PsA ranges from 20 to 670 per 100,000 population [4,5], and 30–50% of PsA patients will develop axial PsA involving the spine or the sacroiliac joints [6]. 

Physical exercise gives beneficial effects against inflammation, joint damage, and symptoms [7] and is recommended as supplementary treatment to patients with arthritis [8]. In addition to positive effects on the joint disease, exercise can reduce the risk for cardiovascular disease (CVD) [9]. High intensity interval training (HIIT) is effective in decreasing CVD risk factors and inflammation [10]. HIIT involves alternating short periods of high intensity exercise with recovery periods or light exercise [11]. Exercise exerts a molecular systemic anti-inflammatory effect related to the intensity and duration of the physical activity [12]. In [13], patients with axial spondyloarthritis experienced reduced disease activity scores and inflammation and improved cardiovascular health after three months of HIIT. Short and long-term beneficial effects of HIIT on disease activity, patient disease perception, and the risk of CVD were recently reported in PsA patients [14,15]. Exercise led to reduced fatigue and cardiovascular risk factors in terms of truncal fat and maximal oxygen uptake, whereas scores for joint affection and pain were compatible with the control group. This finding is important, as there was no detectable negative impact on the disease burden, and HIIT can be recommended for PsA patients. It is, however, possible that vigorous exercise can increase the burden of the joint disease. Bone marrow edema (BME) has been found in relation to physically demanding work and sports activities [16,17,18,19,20]. Mechanical strain can drive inflammatory activity in joints and has been suggested to play a role in the induction and further development of spondyloarthritis [21,22]. 

Magnetic resonance (MR) imaging can portray inflammation in the structures involved [23]. MR-based disease activity scores are reliable and sensitive to change, and they may provide information about disease not given by clinical evaluations [24]. Short tau inversion recovery (STIR) is the recommended MR imaging sequence for axial PsA [25,26], presenting edematous lesions with hyperintense signals [27]. Radiological assessment of edematous lesions is based on hyperintensity in STIR MR images, located in two or more sites and/or two or more slices [28]. Lesions can be confirmed as hypointense signals in T1-weighted MR images. A semi-quantitative scoring system of disease activity, such as the Spondyloarthritis Research Consortium of Canada (SPARCC) scoring system, can be highly sensitive to changes in the spine [26]. Additional information from MR images can be quantified by extracting spatial variations of grey-level intensity in an image [29]. Such a texture analysis may be more sensitive to change than visual inspection and has been applied in studies of BME [30,31,32,33,34,35,36]. It has demonstrated the potential to classify BME from MR images [32,33,34], and to detect changes in bone marrow after physical activity [36]. 

We hypothesized that BME changes could occur in PsA patients after HIIT in spite of no reported changes in disease activity by clinical examinations. The aim of this study was to assess whether HIIT in PsA patients led to detectable changes in the axial skeleton by investigating MR images of the spine for BME. Additionally, we explored the potential of textural features to detect BME changes.

## 2. Materials and Methods

### 2.1. Patient Cohort

The presented study is part of a randomized controlled trial with HIIT as an intervention, conducted at St. Olavs hospital and NTNU—the Norwegian University of Science and Technology, Trondheim, Norway, from 2013 to 2015 [14,15]. Participants fulfilled the ClASsification for Psoriatic ARthritis (CASPAR) criteria [37] and were between 18 and 65 years. Medication histories of participants were collected as has been described previously [14]. The intervention group (N = 19, 4 men, median age 52 years) performed HIIT three times per week for 11 weeks, whereas the control group (N = 20, 8 men, median age 45 years) had no changes from pre-study physical exercise habits. In brief, the intervention was comprised of two weekly supervised stationary bicycling sessions and one weekly self-guided HIIT session. All patients signed informed consent, and the Norwegian Regional Committee for Medical and Health Research Ethics approved the study (Trial registration: NCT02995460). 

### 2.2. Disease Activity Scores

Scores for joint affection and pain were assessed at baseline and after 11 weeks, as previously described [14], and included patient global assessment (PGA), high sensitivity C-reactive protein (hs-CRP), Bath Ankylosing Spondylitis Disease Activity Index (BASDAI), and Disease Activity Score in 44 joints (DAS44). There were no significant differences in baseline scores for joint affection and pain between the intervention and control groups (*p* > 0.05, Wilcoxon signed-rank test).

### 2.3. MR Image Acquisition

MR imaging was performed of the spine in two stations using a STIR and a T1-weighted turbo spin-echo sequence based on a standardized protocol [26] as previously described [38]. For two patients, both in the HIIT group, the second MR imaging was not performed, and both patients were excluded from evaluation by a radiologist and SPARCC scoring, leaving MR images from 37 patients for further analyses. For 10 of the participants, the MR imaging protocol deviated with respect to the spatial resolution in the first (*N* = 1) or second MR imaging (N = 9), and these images were excluded from analysis by textural features. 

### 2.4. Image Analysis

Radiological evaluation

The MR images were evaluated by a radiologist for BME at both time-points. The radiologist was blinded with respect to intervention. BME was identified by hyperintense signals in STIR MR images, supported by hypointense signals in T1-weighted MR images. To be considered positive for BME, the hyperintense signal had to be located in two or more sites and/or two or more slices [28,39]. Images were also assessed with respect to changes from the first to second MR imaging, categorized as stable, increased, or reduced BME. 

SPARCC scoring

The STIR MR images at both time-points were scored by a trained rheumatologist as previously described [26]. In brief, the six most abnormal disco-vertebral levels on the STIR sequence were selected. Three consecutive sagittal slices, which represented the most abnormal slices for each level, were chosen for scoring at that level. The total maximum SPARCC score was 108 for all six levels of the spine. The SPARCC scores were also categorized with respect to changes from the first to second MR imaging, namely stable, increased, or reduced SPARCC score. Categorization to increased or reduced SPARCC score required a minimum score change of five according to the defined minimally important change [40].

Image pre-processing and textural feature extraction

Vertebral bone marrow, excluding vascular and neural structures, were manually segmented using 3D Slicer (MIT Artificial Intelligence Lab, USA) in MR images from all patients (N = 37), comprising images from both time-points (N = 27), or comprising images from only first (N = 9) or only last (N = 1) time-points. Images of the spinal column were pre-processed using a customized intensity adjustment procedure based on the nonparametric nonuniform intensity normalization (N4) bias field correction algorithm [41]. Pixel intensity values were normalized by matching the histogram extracted from the spinal column to the histogram of a randomly selected atlas image. Image noise was reduced using the in-house implementation of an anisotropic (Perona–Malik) diffusion smoothing filter (iterations = 15, integration constant = 1/7, time step = 0.01, conductance = 1.0) [42]. The BME in all image sets was manually segmented using 3D Slicer. The segments were verified by a trained radiologist.

Three pixel-wise types of image textural features were calculated—seven intensity features, ten gradient features, and four grey level co-occurrence matrix (GLCM) textural features, all described in Appendix A. Intensity features were the grey-level intensity values of the central pixel, and the mean, median, standard deviation, minimum, maximum, and semi-interquartile range of the grey-level intensity values. For the extraction of gradient features, 2-dimensional directional gradients for the *x*-axis (G_x) and *y*-axis (G_y) were calculated using a Sobel gradient operator in the imgradientxy function in MATLAB (MathWorks, Natick, MA, USA). GLCM features were extracted using the graycomatrix and graycoprops functions in MATLAB (MathWorks, Natick, MA, USA) at four orientations (0°, 45°, 90°, and 135°) with a distance of 1 pixel. The resulting GLCM feature was the mean of the GLCM feature values in these orientations. Feature maps were created for each feature using a sliding window implementation. In this approach, an orthogonal 3-by-3 box/kernel “slides” in the region of interest, in this case the segmented bone marrow. The features were calculated in each orthogonal kernel position and corresponded to the central pixel of the box. 

### 2.5. Statistical Analysis

Changes in patient characteristics (PGA, hs-CRP, BASDAI, DAS44, and SPARCC score) before and after intervention were analyzed using the Wilcoxon signed-rank test. Changes in BME status and SPARCC score were categorized, and differences between HIIT and the control group were analyzed by Fisher’s exact probability test. Analyses were performed in SPSS (IBM SPSS Statistics v 26), and *p*-values < 0.05 were considered statistically significant. 

Differences in feature values from voxels in BME and healthy voxels were assessed by linear mixed models. We used patient number and scan as random effects, whether the voxel was healthy or pathological as fixed effects, and textural feature as response variables. Further, linear mixed models were used to assess changes between first and second time points and if these changes were different between the HIIT and control groups. For this analysis, average feature values per individual per time point were used as response variables, as it was not possible to match lesions between the two time points. Fixed effects were time (whether the scan was a baseline or 11 week scan), intervention (whether the patient was in the HIIT or the control group), and the interaction term between time and intervention (time*intervention); and patient number was a random effect. The time variable was reference coded to the baseline measurement, and the intervention variable was sum coded. 

Bonferroni correction was used to correct *p*-values for multiple comparisons from all three linear mixed-models. The statistical level of significance was set to *p* < 0.05. Statistical analyses were performed in in RStudio: Integrated Development for R (RStudio, PBC, Boston, MA, USA) and MATLAB R2019A.

## 3. Results

### 3.1. Patient Cohort

Of the 55 patients evaluated for eligibility, 47 were included, and the follow-up at 11 weeks was complete with MR images for 37 patients (Figure 1). Demographic characteristics of study participants in control and HIIT groups, and their scores for joint affection and pain at baseline and after 11 weeks, are shown in Table 1. BASDAI decreased for the HIIT group, and DAS44 decreased for both groups after 11 weeks.

### 3.2. Image Analysis

Radiological evaluation

Examples of acquired MR images are shown in Figure 2, demonstrating the PsA lesions of two patients with low and high disease burden. MR images of the spine from 21 of 37 patients were found negative with respect to BME in the radiological evaluation at both timepoints. Sixteen patients (43%) were identified with BME, consistent with axial manifestation of PsA. The radiologically manifested axial PsA was considered mild to moderate for both groups, and disease burden in terms of BME was stable. The findings are summarized in Table 2. The number of patients with changes of BME after 11 weeks was not significantly different between the HIIT and control group (*p*-value: 0.50).

SPARCC scoring

MR images for 11 of the 37 patients were found negative with respect to BME by the SPARCC scoring at both time-points (Table 3). Twenty-three of the patients (62%) had a positive SPARCC score at both timepoints. Participants in this study had a median baseline SPARCC score of 4.0. The number of patients with changes in SPARCC scores after 11 weeks was not significantly different between the HIIT and control group (*p*-value: 1).

The radiological evaluation and SPARCC scoring were consistent for 49 of the MR images at baseline and 11 weeks. Twenty-three MR images with a positive SPARCC score were BME negative by radiological evaluation. Two MR images with a SPARCC score of 0 were identified as BME positive in the radiological evaluation. SPARCC scoring (Table 3) identified changes from baseline to 11 weeks in more patients than did the radiological evaluation for BME (Table 2).

Textural features

Mean and standard deviation of features extracted from MR images of pathological (BME lesions) and healthy voxels are presented in Table 4. The mean values for all but one (g7) extracted features were significantly different in pathological compared to healthy voxels. With the exception of one of the GLCM features (f1), mean values for textural features were higher in pathological than healthy voxels. We observed no significant differences between the HIIT group and control group in the changes in textural features of PsA lesions from baseline to week 11 (Table 5).

## 4. Discussion

No significant changes were observed in MR images of the spine after HIIT training for 11 weeks. This finding was consistent for radiological evaluation, SPARCC scoring, and textural features of MR images. Values for 20 out of 21 textural features were significantly different in voxels of BME compared to voxels of healthy bone marrow. No textural features of PsA lesions were significantly different when comparing changes in values after 11 weeks between the HIIT and control groups. 

Of the study participants, 43% and 62% were found positive for BME by radiological evaluation and SPARCC scoring, respectively (Table 2 and Table 3). The fraction of BME positive participants by SPARCC scoring is above the reported 30–50% of PsA patients with axial involvement [6]. More positive findings by SPARCC scoring than by the Assessment in SpondyloArthritis international Society criteria have been reported previously [43]. The difference of BME positive participants between the two methods is probably caused by different readers and principal differences in the methods. Standardized methods for scoring of axial spondyloarthritis (axSpA) are subject to some variation between readers [44]. Images of little active inflammation is more subject to low inter-reader correlation, and the mild to moderate disease burden in the cohort of this study is thus suspected to contribute to the difference of the two methods. Both methods rely on hyperintensity in STIR images, where edema related to inflammation can be detected as a bright signal on a dark background in subchondral bone marrow [45]. The use of T1-weighted images to support the radiological evaluation is prone to rejection of positive findings in STIR images, which may explain fewer positive cases by radiological evaluation. The radiological evaluation identified changes in BME from baseline to week 11 for three patients, which agreed with higher and lower SPARCC scores for these patients. SPARCC scoring identified changes from baseline to week 11 for more patients, but in general of minor magnitude. Applying a SPARCC score threshold of five for minimally important change [40] reduced the number of patients with changes from baseline to week 11 from twenty-three to seven. 

Quantitative methods for analysis of STIR MR images have been proven to discriminate between active therapy and placebo after 12 weeks of treatment in clinical trials of ankylosing spondylitis [46,47]. Changes occurring in the spine due to the HIIT should thus be detectable with the current MR imaging protocol and methods for image analysis. MR imaging has previously identified BME in healthy individuals and in athletes, suggesting that mechanical strain contributes to BME [22,43]. Two studies contradict that HIIT may increase disease burden. For this cohort of PsA patients, it has been shown that HIIT has beneficial effects on fatigue and cardiac risk factors without increased joint affection and pain [14,15]. It has also been shown that Ankylosing Spondylitis Disease Activity Score and BASDAI were significantly reduced after 3 months of HIIT in patients with axSpA [13]. Our current study showed no significant changes in BME in the spine from HIIT by radiological evaluation, SPARCC scoring, or texture analysis of MR images (Table 2, Table 3, and Table 5), which supports that HIIT is safe to recommend to patients with PsA.

Mean values of textural features were different in voxels from BME compared to voxels from healthy bone marrow (Table 4). These observed differences are consistent with a previous study, where textural features of MR images have been applied in machine learning to classify active inflammation in sacroiliac joints [35]. Choices of textural features are also important for successful tissue discrimination [30]. We surveyed intensity, gradient, and GLCM textural features, which partly have been utilized in other studies with classification of BME [33,34,48]. These studies also included histogram and run-length matrix features. In studies of osteoarthritis in the knee [34,49,50], most of the texture, histogram, and run-length matrix features were all significantly different between the patient groups. When discriminating the post-radiation lesions edema, fatty conversion, and hemorrhage, Romanos et al. found that GLCM textural features comprised four out of five features in the optimal design of the classification scheme [48]. Sacroiliitis could be classified based on extracted features from STIR MR images and machine learning [35]. The features’ maximum pixel values and LH components from the two-level Haar wavelet decompositions, which depict horizontal traits of the image, were important to discriminate instances. The high intensity gray level in inflammation is a plausible cause for the impact of maximum pixel value for classification of sacroiliitis. Quantitative textural analysis has also been suggested to detect bone structure changes due to exercise [36,51]. This study demonstrated that textural features of BME and non-pathological pixels in STIR MR images were different. 

Limitations of this study include low disease burden in patients and few patients with manifested PsA in the spine. However, if vigorous exercise leads to increased BME, this would be induced also in patients with low disease burden. Furthermore, data on textural features were reduced, since 10 MR image series were excluded due to MRI protocol deviations. However, as all methods for analysis of MR images demonstrated no increase in BME due to HIIT, the indications of no structural changes are fairly strong. To assess the sensitivity to changes of MR image textural features, a longitudinal interventional study of a larger cohort with a higher disease burden would be required.

The thorough evaluation of spine MR images demonstrate that the PsA is stable regarding BME in the spine under 11 weeks of HIIT. Beneficial effects of HIIT in this patient cohort have been previously reported, with increased maximal oxygen consumption (VO2max), reduced truncal fat, and less fatigue [14,15]. Importantly, joint affection and pain did not differ from the control group. The negative findings also in MR images strongly indicate no structural changes. The evidence of HIIT being safe to conduct for patients with PsA is thus stronger. 

## Figures and Tables

**Figure 1 diagnostics-12-01420-f001:**
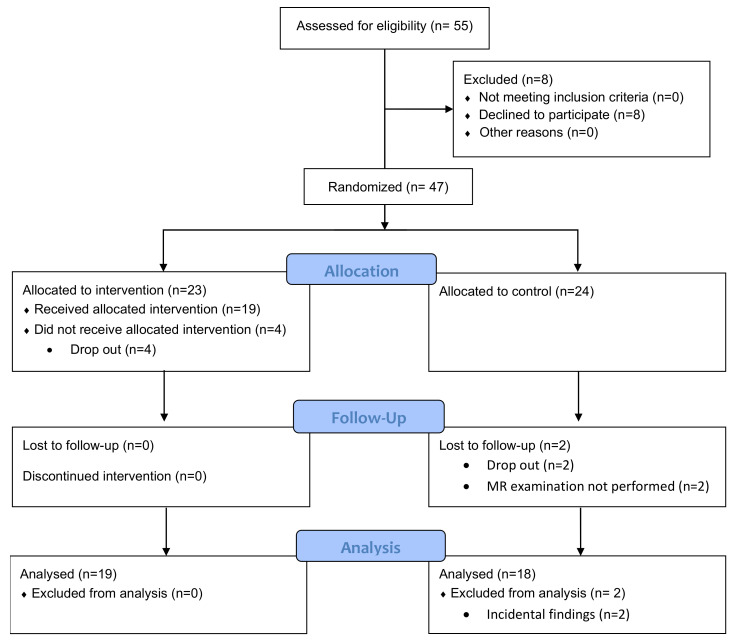
Flow chart of patient inclusion, follow-up, and analysis.

**Figure 2 diagnostics-12-01420-f002:**
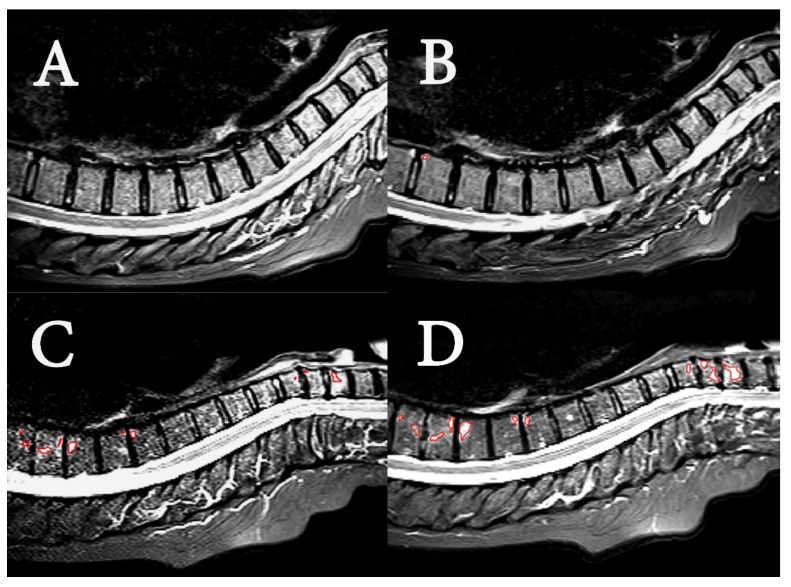
MR image examples. Sagittal short tau inversion recovery (STIR) MR images of upper spine in two patients. (**A**) (baseline) and (**B**) (11 weeks) are of a patient with low SPARCC scores (4 and 2, respectively). The images show the spine from the second cervical vertebrae (C2) to thoracic vertebrae 12 (TH12). Segmented region in red in B shows minor BME in corner of TH10. (**C**) (baseline) and (**D**) (11 weeks) are of patient with high SPARCC scores (39 and 50, respectively). The images show the spine from the second cervical vertebrae (C2) to thoracic vertebrae 11 (TH11), and segmented regions in red in (**C**) and (**D**) show BME in C5 to C7, and in TH5 to TH9.

**Table 1 diagnostics-12-01420-t001:** Demographic characteristics of study participants in high intensity interval training (HIIT) and control groups, and their scores for joint affection and pain at baseline and after 11 weeks [15]. Mean values of scores for joint affection and pain within groups at baseline and 11 weeks were compared by the Wilcoxon signed-rank test.

	HIIT			Controls		
	Baseline	11 Weeks	*p*	Baseline	11 Weeks	*p*
Number, n	19		-	20		-
Men, n (%)	4 (21)		-	8 (40)		-
Women, n (%)	15 (79)		-	12 (60)		-
Age, median (range)	52 (39–64)		-	45 (23–64)		-
PGA ^a^, median (range)	50.0 (1.0–95.0)	43.0 (3.0– 81.0)	0.465	46.0 (6–85)	35.5 (0–89)	0.227
hs-CRP ^b^ mg/L, median (range)	1.8 (0.4–24.0)	2.1 (0.5–10.2)	0.096	2.2 (0.1–28.7)	2.2 (0.3–22.0)	0.571
BASDAI ^c^, median (range)	4.0 (0.4–8.3)	3.2 (0.5–6.6)	0.049	3.7 (0.3–6.7)	2.6 (0.2–7.7)	0.133
DAS44 ^d^, median (range)	2.3 (0.8–3.3)	1.9 (0.5–2.4)	0.001	2.3 (0.6–3.1)	1.7 (0.6–3.0)	0.007

^a^ PGA: Patient Global Assessment, range from 0 to 100. ^b^ hs-CRP: high-sensitive C-reactive protein. ^c^ BASDAI: Bath Ankylosing Spondylitis Disease Activity Index, range from 0 to 10. ^d^ DAS44: disease activity score in 44 joints, range from 0.2 to 9.9.

**Table 2 diagnostics-12-01420-t002:** Bone marrow edema by radiological evaluation of MR images. Results from radiological evaluation for bone marrow edema (BME) in short tau inversion recovery (STIR) and T1-weighted MR images of the participants in high intensity interval training (HIIT) and control groups at baseline and after 11 weeks. Number of participants with detectable changes after 11 weeks were not significantly different for the two groups (Fisher’s Exact Probability Test *p*-value: 0.50).

	HIIT		Controls	
	Baseline	11 Weeks	Baseline	11 Weeks
Number, n	17		20	
BME detected, n (%)	9 (53)	9 (53)	5 (25)	5 (25)
No change, n (%)	17 (100)		17 (85)	
Increased BME, n (%)	0 (0)		1 (5)	
Reduced BME, n (%)	0 (0)		2 (10)	

**Table 3 diagnostics-12-01420-t003:** Results from evaluation of short tau inversion recovery MR images of the participants in high intensity interval training (HIIT) and controls groups at baseline and after 11 weeks by SPARCC scoring [26]. If SPARCC score changed by five or more to week 11, it was categorized as increased or decreased according to minimally important changes [40]. Number of participants with detectable changes after 11 weeks were not significantly different for the two groups (Fisher’s Exact Probability Test *p*-value: 1).

	HIIT		Controls	
	Baseline	11 Weeks	Baseline	11 Weeks
Number, n	17		20	
SPARCC ^a^ score > 0, n (%)	13 (72)	13 (72)	13 (65)	10 (50)
SPARCC ^a^ score, median (max value)	4.0 (39)	5.0 (50)	4.0 (36)	0.5 (20)
No change in SPARCC ^a^ score, n (%)	14 (82)		16 (80)	
Increased SPARCC ^a^ score, n (%)	1 (6)		1 (5)	
Reduced SPARCC ^a^ score, n (%)	2 (12)		3 (15)	

^a^ SPARCC: SpondyloArthritis Research Consortium of Canada, score range from 0 to 108.

**Table 4 diagnostics-12-01420-t004:** Textural features of voxels in bone marrow edema and healthy voxels. Mean values and standard deviation (SD) of features extracted from MR images of voxels in bone marrow edema (BME) and healthy voxels. Differences in feature values for pathological and healthy voxels were examined using linear mixed-effects models. *p*-values and Bonferroni corrected *p*-values (q-values) are reported. Significant values (<0.05) after Bonferroni in bold.

	Voxels with BME (N = 3289)		Healthy Voxels (N = 3289)			
	Mean	*SD*	Mean	*SD*	*p*-Value	*q*-Value
i_1_	159.8	21.0	117.1	21.7	<0.001	**<0.001**
i_2_	150.1	18.0	116.6	16.9	<0.001	**<0.001**
i_3_	153.2	19.0	117.5	16.5	<0.001	**<0.001**
i_4_	20.6	10.2	13.4	7.3	<0.001	**<0.001**
i_5_	104.3	34.2	87.8	29.8	<0.001	**<0.001**
i_6_	178.2	18.4	139.9	16.7	<0.001	**<0.001**
i_7_	14.1	8.2	8.6	5.2	<0.001	**<0.001**
g_1_	3219.6	1571.4	2231.4	1188.8	<0.001	**<0.001**
g_2_	2557.3	1250.6	1777.9	948.2	<0.001	**<0.001**
g_3_	128.8	62.9	89.3	47.6	<0.001	**<0.001**
g_4_	102.3	50.0	71.1	38.0	<0.001	**<0.001**
g_5_	78.1	43.3	51.4	31.5	<0.001	**<0.001**
g_6_	61.5	34.0	40.9	25.1	<0.001	**<0.001**
g_7_	115.5	59.9	80.4	43.9	<0.001	**<0.001**
g_8_	20.3	17.8	15.9	12.8	0.062	1
g_9_	292.2	150.5	201.0	115.1	<0.001	**<0.001**
g_10_	56.3	34.4	36.2	23.6	<0.001	**<0.001**
f_1_	4.52	1.7	5.31	1.91	<0.001	**<0.001**
f_2_	0.52	0.23	0.32	0.22	<0.001	**<0.001**
f_3_	0.13	0.09	0.09	0.03	<0.001	**<0.001**
f_4_	0.56	0.08	0.51	0.06	<0.001	**<0.001**

**Table 5 diagnostics-12-01420-t005:** Mean values and standard deviation (SD) of features extracted from BME lesions in MR images of psoriatic arthritis patients. Differences between the intervention and control group, in terms of changes in feature values from the baseline to the 3-month scan, were investigated using a linear mixed effect model (random effects: patient number and lesion number (nested effects), fixed effects: baseline or 3 months scan and control or intervention group). *p* values are reported.

	Intervention (N = 1174)		Control (N = 2115)		
	Mean	*SD*	Mean	*SD*	*p*
i_1_	160.6	22.2	159.3	20.4	0.56
i_2_	151.9	20.1	149.2	16.6	0.71
i_3_	155.4	22.2	152.0	16.8	0.74
i_4_	19.4	9.4	21.2	10.6	0.84
i_5_	107.6	34.0	102.5	34.1	0.65
i_6_	176.4	15.8	179.2	19.6	0.73
i_7_	13.2	8.4	14.7	8.0	0.92
g_1_	2956.9	1439.8	3365.4	1621.9	0.87
g_2_	2346.1	1141.0	2674.5	1293.0	0.87
g_3_	118.3	57.6	134.6	64.9	0.87
g_4_	93.8	45.6	107.0	51.7	0.87
g_5_	77.8	42.0	78.2	44.0	0.82
g_6_	61.0	32.4	61.8	34.8	0.81
g_7_	103.9	57.7	121.9	60.2	0.82
g_8_	15.9	15.5	22.8	18.5	0.95
g_9_	283.7	141.2	297.0	155.2	0.79
g_10_	55.5	33.9	56.7	34.6	0.95
f_1_	4.73	1.84	4.40	1.60	0.75
f_2_	0.50	0.23	0.53	0.23	0.72
f_3_	0.16	0.13	0.11	0.04	0.23
f_4_	0.59	0.10	0.54	0.06	0.26

## Data Availability

Data are not available, as ethical approval of project does not include transfer of data.

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
