# Peer review of "Evaluating the Impact of High Intensity Interval Training on Axial Psoriatic Arthritis Based on MR Images"

_diagnostics, 2022, doi:10.3390/diagnostics12061420_

Round 1

Reviewer 1 Report

I read with great interest the article titled "Evaluating the impact of high intensity interval training on ax-2 ial psoriatic arthritis based on MR images" by Chronaiou etcolleagues.

ABSTRACT

Please add more numerical results since your research article is rich

INTRODUCTION

Please add a sentence where authors describe psoriasis as a chronic inflammatory disease with potential joints manifestations (psoriatic arthritis) and also add the Global Burden of Diseases most recent epidemiological study published.

Please explain that physical exercise is useful for PsA patients and may help to decrease cardiovascular [10.3390/jcm9010186] and respiratory inflammation [10.1007/s10067-020-05050-2]

Methods

Please cite Caspar reference

Results

Table 1: Please change years with age

Discussion

Please discuss how SARS-CoV-2 may be able to trigger PsA flares [10.3389/fimmu.2021.635018] and how important practicing sport will become in the present and in the future.

Author Response

ABSTRACT

Please add more numerical results since your research article is rich

We acknowledge this comment from reviewer. We have considered to add numerical results of the three methods for analysis of MR images after intervention. This would provide numbers that demonstrate no significant difference between HIIT and control groups after 11 weeks. Results from textural features are difficult to present briefly by numbers. Thus, we decided to keep these numbers out of Abstract, as we found it more important that the main finding is clearly stated: that HIIT leads to no change in bone marrow edema.

INTRODUCTION

Please add a sentence where authors describe psoriasis as a chronic inflammatory disease with potential joints manifestations (psoriatic arthritis) and also add the Global Burden of Diseases most recent epidemiological study published.

We are a bit unsure of this comment. The first sentence of the Introduction says: Psoriatic Arthritis (PsA) is a chronic inflammatory joint disease associated with skin psoriasis…… We believe this contains what reviewer asks, but with PsA as starting point. And when it comes to publications from the Global Burden of Diseases, we were not able to find any relevant recent literature that includes PsA.

Please explain that physical exercise is useful for PsA patients and may help to decrease cardiovascular [10.3390/jcm9010186] and respiratory inflammation [10.1007/s10067-020-05050-2]

We have rewritten parts of the introduction (lines 38 – 50) due to comments on this from reviewers.

Methods

Please cite Caspar reference.

Thank you for pointing this out. CASPAR is now cited (line 85).

Results

Table 1: Please change years with age.

This is now done (page 6).

Discussion

Please discuss how SARS-CoV-2 may be able to trigger PsA flares [10.3389/fimmu.2021.635018] and how important practicing sport will become in the present and in the future.

Reviewer points to an important issue, that infections by viruses may worsen the health for PsA patients and that physical exercise can be important to prevent this. It has already been shown that vigorous physical exercise leads to no change in the clinical evaluation for this patient group. The focus for this paper was to strengthen this finding, by showing that there was no underlying radiographic change either. We therefore think that including SARS-CoV-2 will complicate the discussion. We have not discussed overweight for the same reason.

Reviewer 2 Report

Thank you very much for allowing me to review this manuscript entitled “Evaluating the impact of high-intensity interval training on axial psoriatic arthritis based on MR images”. Unfortunately, I think this manuscript could not be published unless a substantial revision. There are many aspects of this study that need to be better argued. Below I describe some points that I think the authors should address so that this study could be published in this journal.

Abstract
The author should illustrate the design of the participants' distribution, especially the randomization and the blinding methods used in the trial, as in Line 23-24.
The author should illustrate the basic information of the HIIT protocol such as movements, intensity, and frequency, as in Line 23-24.

Introduction
Line 44-45, the author mentioned the effect of HIIT on inflammation for patients with PsA, however, it is confusing that the author mentioned the beneficial effects of HIIT on disease activity, patient disease perception, and the risk of cardiovascular disease. What is the logical connection between them?
Line 46-47, why did the authors say “exercise reduces fatigue” (exercise could also induce fatigue) and mentioned cardiovascular risk factors?
Line 49, the authors didn’t clarify the definition of HIIT.
Line 41-53, the authors need to establish a reasonable logical correlation between HIIT and PsA.
In a clinical trial, the BME investigated by MRI and the SPARCC were the outcome measures, however, in clinical practice, DAS-28, MDA, RAPID, and DAPSA were mostly used in the clinical assessment of PsA. I doubt that the outcome measures in the trial would not be strong enough to identify the safety of HIIT. The author should demonstrate the reliability of the choice of BME.

Materials and Methods
Line 79, the authors mentioned that the control group did no change in pre-study physical exercise habits, I doubt it might induce heterogeneity between groups. However, the authors didn’t compare the characteristics between groups in the baseline.
The author should illustrate the information of the HIIT protocol.
The authors did not collect the medication history of the participants.
Since the author did not strictly control variables within and between groups, the interference of covariables should be excluded from the statistical analysis. At the same time, considering the small sample size of the study, it is suggested that the author conduct a statistical power analysis.

Results
As a manuscript of a clinical intervention trial, the authors should provide results more formally. I suggested formatting this manuscript according to SPIRIT or CONSORT statements.
The authors didn’t provide the rate of loss of follow-up and incidence of adverse events.

Discussion
According to the definition of HIIT that it is a training protocol alternating short periods of intense or explosive anaerobic exercise with brief recovery periods until the point of exhaustion, which thereby relies on the anaerobic energy releasing system almost maximally. Therefore, HIIT is considered a form of cardiovascular exercise. However, PsA is an inflammatory musculoskeletal disease associated with psoriasis. The author should clarify the potential mechanism of the functioning of HIIT in PsA treatment, which is also an important logical basis for the study.
As in Line 262-273, the authors paid too much attention to illustrating their outcome measures. As mentioned above, if the authors want to demonstrate the safety of HIIT in the treatment of PsA patients, they should clarify the potential risk of HIIT within the PsA population and provide clinical evidence.
The authors didn’t discuss the limitation of this study. 

Author Response

Thank you very much for allowing me to review this manuscript entitled “Evaluating the impact of high-intensity interval training on axial psoriatic arthritis based on MR images”. Unfortunately, I think this manuscript could not be published unless a substantial revision. There are many aspects of this study that need to be better argued. Below I describe some points that I think the authors should address so that this study could be published in this journal.

We thank the reviewer for thorough evaluation and feedback of the manuscript. We have revised it according to the issued points, although not completely in line with all comments from reviewer. In particular, we have not restructured the manuscript according to SPIRIT or CONSORT statements. Our key reason for this is to keep the manuscript as brief and to the point as possible. This is a study on secondary outputs on a part of the cohort in a clinical trial, so primary outputs have been reported previously (Thomsen 2018 and 2019). These papers follow the guidelines for RCT reporting, and contain detailed information on randomization, blinding, and so on.

Abstract
The author should illustrate the design of the participants' distribution, especially the randomization and the blinding methods used in the trial, as in Line 23-24.

Randomization and blinding are reported in previous publications (Thomsen 2018 and 2019).

The author should illustrate the basic information of the HIIT protocol such as movements, intensity, and frequency, as in Line 23-24

We have added a very brief description, and refer to previous publication (Thomsen 2018).

Introduction

Line 44-45, the author mentioned the effect of HIIT on inflammation for patients with PsA, however, it is confusing that the author mentioned the beneficial effects of HIIT on disease activity, patient disease perception, and the risk of cardiovascular disease. What is the logical connection between them?

We have tried to make these connections clearer, and this part of the manuscript has been rewritten. In brief; PsA is associated with several comorbidities, and cardiovascular disease is one. and we have put more emphasis on the expected benefits on CVD. We hope reviewer now find that the connection is more logical.

Line 46-47, why did the authors say “exercise reduces fatigue” (exercise could also induce fatigue) and mentioned cardiovascular risk factors?

HIIT was found to reduce fatigue in the referred study, of the cohort which these patients are part of (Thomsen 2019).

Line 49, the authors didn’t clarify the definition of HIIT.

A description of HIIT has been added.

Line 41-53, the authors need to establish a reasonable logical correlation between HIIT and PsA.

Patients with PsA are expected to have several benefits from HIIT. Particularly risk factors for CVD should be reduced. It could potentially also decrease inflammation. This part of the introduction have been rewritten due to reviewer comments, and we have put more emphasis on the expected benefits on CVD. We hope reviewer now find that the correlation between HIIT and PsA is more logical.

In a clinical trial, the BME investigated by MRI and the SPARCC were the outcome measures, however, in clinical practice, DAS-28, MDA, RAPID, and DAPSA were mostly used in the clinical assessment of PsA. I doubt that the outcome measures in the trial would not be strong enough to identify the safety of HIIT. The author should demonstrate the reliability of the choice of BME.

HIIT has previously been demonstrated to be safe for PsA patients in terms of clinical assessment, including PGA, fatigue, DAS44, pain, ASDAS-CRP and hs-CRP (Thomsen 2019). The main purpose of this manuscript was to further strengthen the evidence of that HIIT is safe for PsA patients by demonstrating no change in BME.

Materials and Methods
Line 79, the authors mentioned that the control group did no change in pre-study physical exercise habits, I doubt it might induce heterogeneity between groups. However, the authors didn’t compare the characteristics between groups in the baseline.

We have now included statistical test of baseline values between intervention and control groups that show that there are no differences (lines 98 - 106).

The author should illustrate the information of the HIIT protocol.

We are a bit unsure of this comment as reviewer has commented on the HIIT protocol previously. We thus believe our answer to previous comment should apply to this as well: We have added a very brief description, and refer to previous publication (Thomsen 2018).

The authors did not collect the medication history of the participants.

Thank you for pointing this out. It was recorded and has been extensively described previously. We now refer to the publication that describes this.

Since the author did not strictly control variables within and between groups, the interference of covariables should be excluded from the statistical analysis. At the same time, considering the small sample size of the study, it is suggested that the author conduct a statistical power analysis.

We are not sure what reviewer means with excluding interference of covariables. This was an RCT, which is why variables within and between groups were not controlled. We have applied linear mixed models, with the interaction between intervention and time in the model. This is a valid and recommended approach for analysis of RCT data (Twisk 2018; doi: 10.1016/j.conctc.2018.03.008).

The statisticians at our department strongly recommend that we don’t perform a statistical power analysis after the experiments have been done, as this would provide no additional information to the p-values (Lydersen 2019; doi: 10.4045/tidsskr.18.0847), thus we have not performed a post hoc power calculation in this study.

Results
As a manuscript of a clinical intervention trial, the authors should provide results more formally. I suggested formatting this manuscript according to SPIRIT or CONSORT statements.

This manuscript describes analysis performed on a sub-cohort of an RCT where 67 patients were included in total. In the current manuscript, we present findings by MRI, which was performed on approximately 2/3 of the cohort. The previous publications (Thomsen 2018 and 2019) are in accordance with RCT reporting standards. We have added a flow-chart following the CONSORT standard, which should answer some of reviewers questions. Otherwise, we have included as much of the requested information in CONSORT guidelines as doable for this sub-cohort, whereas lacking information can be found in the referred open access publications for the full cohort (Thomsen 2018 and 2019).

The authors didn’t provide the rate of loss of follow-up and incidence of adverse events.

We have now added a flowchart by the CONSORT standards, new Figure 1, that show this.

Discussion
According to the definition of HIIT that it is a training protocol alternating short periods of intense or explosive anaerobic exercise with brief recovery periods until the point of exhaustion, which thereby relies on the anaerobic energy releasing system almost maximally. Therefore, HIIT is considered a form of cardiovascular exercise. However, PsA is an inflammatory musculoskeletal disease associated with psoriasis. The author should clarify the potential mechanism of the functioning of HIIT in PsA treatment, which is also an important logical basis for the study.

We believe this comment is related to previous ones and think that this comment also should be answered by the revised introduction.

As in Line 262-273, the authors paid too much attention to illustrating their outcome measures. As mentioned above, if the authors want to demonstrate the safety of HIIT in the treatment of PsA patients, they should clarify the potential risk of HIIT within the PsA population and provide clinical evidence.

We believe this comment is simalr to a previous, and thus use the same answer: HIIT has previously been demonstrated to be safe for PsA patients in terms of clinical assessment, including PGA, fatigue, DAS44, pain, ASDAS-CRP and hs-CRP (Thomsen 2019). The main purpose of this manuscript was to further strengthen the evidence of that HIIT is safe for PsA patients by demonstrating no change in BME.

The authors didn’t discuss the limitation of this study. 

Thank you for this comment. We have now added a discussion of limitations (lines 426 – 431).

Round 2

Reviewer 2 Report

The authors are to be commended for their significant work in revising this manuscript. There are substantial changes that address most of the comments, with great improvement in scope and clarity. However, there are still few issues need to be noticed. Some suggestions are listed in the specific comments below

Specific comments:

1.     In the abstract part, line 23-24, “PsA patients went through 11 weeks of HIIT (N=19) or no change in physical exercise habits (N=20).” It is recommended to provide detailed anthropometry information for patient, such as gender, height, weight.

2.     In the abstract part, line 24-25, “We acquired scores for joint affection and pain and STIR and T1-weighted MR images of the spine at both timepoints.” What are two timepoints? Can you provide more details?

3.     In the abstract part, please provide a brief description of the implications and highlight of this paper.

4.     In the introduction part, line 47-50, “HIIT involves……after three months of HIIT” please improve the logical coherence between these two sentences.

5.     In the last paragraph of introduction part, it is recommended to add the hypothesis of this manuscript.

6.     In the materials and methods part, is the sample size appropriate?

7.     In the materials and methods part, MR image acquisition, line 110, “an inversion recovery-based sequence (STIR)” The abbreviation “STIR” has appeared previously, please consider deleting the illustration here.

8.     In the discussion part, it is recommended to provide a brief description of the aim and main findings in the first paragraph of the manuscript.

9.     In the discussion part, line 375, “SpondyloArthritis international Society (ASAS) criteria…” please delete the abbreviation “ASAS”, since it just appeared once in the article. There are some unnecessary abbreviations in this manuscript, please check this problem.

10.  It is recommended to add a conclusion that shows detailed findings about this manuscript as well as what are the contributions for future clinical or scientific research.

Author Response

We thank reviewer for thorough and detailed input for improvements of the manuscript. Reviewer found that further improvements were needed and have generously provided a detailed list that can help sort these out. We have taken almost all suggestions into account, and hope that reviewer and editor will find the the current manuscript of acceptable quality for publication.  

  1. In the abstract part, line 23-24, “PsA patients went through 11 weeks of HIIT (N=19) or no change in physical exercise habits (N=20).” It is recommended to provide detailed anthropometry information for patient, such as gender, height, weight.

We have included gender and age, in abstract and under Materials and Methods

  1. In the abstract part, line 24-25, “We acquired scores for joint affection and pain and STIR and T1-weighted MR images of the spine at both timepoints.” What are two timepoints? Can you provide more details?

      This was at baseline and after 11 weeks. “both timepoints” have been replaced with this information.

  1. In the abstract part, please provide a brief description of the implications and highlight of this paper.

      Here we do not understand what reviewer requires. Main finding and its implication is stated as the conclusion: “BME in spine was not changed after HIIT, supporting that HIIT is safe for PsA patients.

  1. In the introduction part, line 47-50, “HIIT involves……after three months of HIIT” please improve the logical coherence between these two sentences.

We have added this statement, with reference to a review by Gleeson: «Exercise exerts a molecular systemic anti-inflammatory effect, related to the intensity and duration of the physical activity»

  1. In the last paragraph of introduction part, it is recommended to add the hypothesis of this manuscript.

We have included the hypothesis that that BME changes could occur in PsA patients after HIIT, in spite of no reported change in disease activity by clinical examinations

  1. In the materials and methods part, is the sample size appropriate?

We think so. As finding of this study is negative, the appropriate sample size is difficult to decide. A larger sample size would strengthen the evidence, but would also demand substantial resources. This study aimed to support previous findings of HIIT for PsA patients being safe, and MR images have been thoroughly investigated with this in mind.

  1. In the materials and methods part, MR image acquisition, line 110, “an inversion recovery-based sequence (STIR)” The abbreviation “STIR” has appeared previously, please consider deleting the illustration here.

      We followed this suggestion and changed to STIR.

  1. In the discussion part, it is recommended to provide a brief description of the aim and main findings in the first paragraph of the manuscript.

      Author guidelines for Diagnostics do not contain this specific direction for the Discussion (“Authors should discuss the results and how they can be interpreted in perspective of previous studies and of the working hypotheses.The findings and their implications should be discussed in the broadest context possible and limitations of the work highlighted. Future research directions may also be mentioned”). We want to avoid repeating the aim here. We do agree with reviewer that the Discussion should start with main findings, which it does.

  1. In the discussion part, line 375, “SpondyloArthritis international Society (ASAS) criteria…” please delete the abbreviation “ASAS”, since it just appeared once in the article. There are some unnecessary abbreviations in this manuscript, please check this problem.

      We followed this suggestion and removed the abbreviation ASAS. We have also checked and edited the manuscript for further unnecessary abbreviations.

  1. It is recommended to add a conclusion that shows detailed findings about this manuscript as well as what are the contributions for future clinical or scientific research.

Manuscript guidelines for Diagnostics says about Conclusions: “This section is not mandatory but can be added to the manuscript if the discussion is unusually long or complex.”

As the Discussion is rather shot (78 lines), we find the current presentation to be in line with journals guidelines.

Round 3

Reviewer 2 Report

This study try to support that HIIT for PsA patients being safe, and MR images have been thoroughly investigated to concrete that idea. Authors have made a good revision, I agree to accept it.